# Beetle Antennae Search: Using Biomimetic Foraging Behaviour of Beetles to Fool a Well-Trained Neuro-Intelligent System

**DOI:** 10.3390/biomimetics7030084

**Published:** 2022-06-23

**Authors:** Ameer Hamza Khan, Xinwei Cao, Bin Xu, Shuai Li

**Affiliations:** 1Smart City Research Institute, The Hong Kong Polytechnic University, Kowloon 999077, Hong Kong; ahakhan@polyu.edu.hk; 2School of Business, Jiangnan University, Wuxi 214122, China; 3School of Automation, Northwestern Polytechnical University, Xi’an 710072, China; mileface.binxu@gmail.com; 4School of Informatics, Lanzhou University, Lanzhou 730000, China; lishuai@lzu.edu.cn

**Keywords:** fooling attacks, nature-inspired algorithm, cognitive intelligence, neuro-intelligent systems

## Abstract

Deep Convolutional Neural Networks (CNNs) represent the state-of-the-art artificially intelligent computing models for image classification. The advanced cognition and pattern recognition abilities possessed by humans are ascribed to the intricate and complex neurological connection in human brains. CNNs are inspired by the neurological structure of the human brain and show performance at par with humans in image recognition and classification tasks. On the lower extreme of the neurological complexity spectrum lie small organisms such as insects and worms, with simple brain structures and limited cognition abilities, pattern recognition, and intelligent decision-making abilities. However, billions of years of evolution guided by natural selection have imparted basic survival instincts, which appear as an “intelligent behavior”. In this paper, we put forward the evidence that a simple algorithm inspired by the behavior of a beetle (an insect) can fool CNNs in image classification tasks by just perturbing a single pixel. The proposed algorithm accomplishes this in a computationally efficient manner as compared to the other adversarial attacking algorithms proposed in the literature. The novel feature of the proposed algorithm as compared to other metaheuristics approaches for fooling a neural network, is that it mimics the behavior of a single beetle and requires fewer search particles. On the contrary, other metaheuristic algorithms rely on the social or swarming behavior of the organisms, requiring a large population of search particles. We evaluated the performance of the proposed algorithm on LeNet-5 and ResNet architecture using the CIFAR-10 dataset. The results show a high success rate for the proposed algorithms. The proposed strategy raises a concern about the robustness and security aspects of artificially intelligent learning systems.

## 1. Introduction

The level of intelligence, learning, and cognitive ability possessed by living beings is directly associated with the structural complexity [1,2] and the size of the brain [3,4,5]. Human brains demonstrate an excellent ability to discover, learn, and recognize intricate patterns because of complex interconnection between neurons [6]. The complexity of the neural structure of the human brain has inspired the development of artificially intelligent learning systems [7,8,9,10,11], which are rapidly coming at par with the performance of human brain itself [12,13,14]. As shown in Figure 1, on the other end of the biological intelligence spectrum, lies small organisms e.g., worms and insects, which demonstrate an rudimentary-intelligent behavior and limited learning ability [15] because the neurological structure is not complex enough to develop intelligent reasoning. However, in this paper, we demonstrate a counter-intuitive result that an artificial learning system, inspired by the human brain, can be fooled by following a straightforward algorithm inspired by the behavior of an insect. More specifically, the proposed algorithm is inspired by the Beetle Antennae Search (BAS) algorithm based on the food foraging behavior of beetles [16,17,18,19]. This simple algorithm can successfully fool a deliberately designed and extensively trained Convolutional Neural Networks (CNNs) to make the wrong decisions for recognizing objects in input images. We anticipate that the presented results will provide further insight into the nature of artificial learning systems. Furthermore, these results create important implications for the security and robustness of the intelligent computation models and how it will affect the future of artificial intelligence, particularly since such systems are being widely used in commercial products [20,21,22,23,24].

The proposed algorithm belongs to the class of nature-inspired metaheuristic algorithms. In recent years, nature-inspired optimization has been an active area of research and has given rise to several efficient optimization algorithms [11,19,25]. For example, biological evolution and natural selection have inspired the formulation of Evolutionary and Genetic Algorithms (GAs) [26]. Similarly, the flocking and swarming behavior (also referred to as swarm intelligence) of birds has inspired the creation of Particle Swarm Optimization (PSO). Similarly several other algorithms have been proposed inspired by social behaviour of other animals and insects, e.g., Ant Colony Optimization (ACO) [27], Artificial Fish Swarm Algorithm [28], Cuckoo Search [29], Invasive Weed Optimization (IWO) [30], Honey Bee Algorithm (HBA) [31], and Firefly Algorithms (FAs) [32], Grey Wolf Optimizer (GWO) [33], Ref Fox Optimizer (RFO) [34,35], and Black Widow Optimization (BWO) [36,37]. The interesting difference between BAS and other nature-inspired algorithms is that it isn’t based on the swarming or social behavior of the animals. It is inspired by the food foraging behavior of a single beetle. Beetle behavior is of particular research interest because, unlike other insects, beetles usually do not work in a swarm and have the ability to search for food individually. It requires fewer search particles and objective function evaluations in each iteration of the algorithm, making it computationally efficient. Table 1 shows comparison between different metaheuristic algorithms proposed in literature for fooling neural network models.

Deep CNNs are a special type of neural network, designed specifically for learning the features from images. CNNs make use of spatial correlation of image pixels and therefore match the mechanism of how the human brain process visual signal. CNNs have been an active area of research in recent years [14,42,43] and are able to outperform the traditional processing algorithms in several tasks such as, image classification [44], segmentation [45], and restoration [46]. However, several studies also approached the CNNs from the perspective of security, robustness, and sensitivity to the noise in the input image. Szegedy et al. [47] highlight the sensitivity of the CNN to different factors of the input image, e.g., colors, light direction, the posture of objects, and several other factors. Based on this, several algorithms have been proposed [48] to fool CNN to misclassify an input image. Most of these algorithms work by adding a properly designed perturbation to the input image such that the CNN will eventually misclassify it. Goodfellow et al. [41,48,49,50] proposed a mechanism to calculate the required perturbation to fool a CNN by exploiting their linear behavior in high-dimensional space. Similarly, Moosavi-Dezfooli et al. [49] proposed DeepFool, an optimization-based algorithm to find the required perturbation. Su et al. [41] extended the scope of attacking strategies by fooling a CNN by perturbing just a single pixel. They used a population-based metaheuristic algorithm, called differential evolution, to find an optimal image perturbation. However, these strategies either require the estimation of the gradient of the objective function or use a population of several searching particles, which make them computationally expensive to execute.

In this paper, we propose a computationally efficient searching strategy based on the nature-inspired food foraging behavior of beetle to fool a CNN by just modifying a single pixel. Nature-inspired optimization algorithms [51,52,53] have been popular in the literature because of their ability to solve complex nonlinear optimization problems without relying on the gradient of the objective function [54,55,56,57,58]. They show efficient computational properties and fast convergence performance [59,60,61,62]. First, we mathematically formulate the CNN fooling as a constrained optimization problem. The objective function takes the pixel location, and brightness value as input, and its value is proportional to the confidence of CNN in the real class. The solution to the optimization problem will minimize the confidence of CNN in a real class, such that it will output the wrong class. Next, we propose the optimization algorithm, which uses a single search particle to search for the pixel-perturbation, to fool the CNN.

It should be noted that in contrast to the current algorithm, the proposed algorithm neither requires the gradient of the objective function nor the use of a population of several search particles. In fact, it just uses a single search particle, which makes it much more numerically efficient for a large-scale attack on a CNN. This work is motivated by the need to highlight the weakness of artificial learning systems. Our aim is to motivate the development of intelligent computation models with robustness as a primary concern in addition to accuracy [43]. The main highlights of this paper are listed as follows:A pixel-level fooling attack algorithm for CNN, by just using a single search particle.The algorithm is independent of the architecture of the CNN. It treats the CNN as a black-box and just relies on the output prediction of the CNN. Therefore, the algorithm is general enough to be applied to different CNN architectures.The algorithm is very efficient since it relies on only a single search particle.Extensive experimental results using two CNN architectures; LeNet-5 and ResNet are presented to demonstrate the efficacy of the proposed algorithm.

The rest of the paper is organized as follows. Section 2 will present the problem formulated and mathematically model the CNN fooling as a constrained optimization problem. Section 3 will formulate the algorithm to solve the formulated optimization problem. Section 4 presents the methodology to evaluate the performance of the proposed algorithm, along with the experimental results. Section 5 concludes the paper.

## 2. Problem Formulation

In this section, we will present the mathematical formulation of the optimization problem used to conduct a pixel-level attack on the CNN. First, we briefly describe the structure of CNNs, then discuss two types of attacks; targeted and untargeted.

### 2.1. Convolutional Neural Networks

CNNs belong to a special type of neural networks, designed specifically for learning the features on visual data, e.g., images. The architecture of CNNs allows taking advantage of spatial correlation between a neighboring pixel of the images, a feature lacking from a simple feedforward neural network. In CNNs, the input of the network is followed by convolutional layers and downsampling layers, before the fully connected layers, to extract useful image features efficiently. A typical CNN is shown in Figure 2. It is a simple CNN architecture with a convolutional layer, a downsampling layer, and a fully connected layer followed by a softmax layer to output probabilities of the image belonging to a particular class.

The most common application of CNN is image classification. It takes an image as the input, and assign probabilities to each output class. The probabilities represent the confidence of the neural network, whether the input image belongs to a particular class. To write it mathematically, a CNN can be considered a nonlinear function $f_{cnn}$ applied on an image input $X_{img}$, which return the probability for each class
(1)$$\begin{array}{c} p=f_{cnn}\left(X_{img}\right), \end{array}$$
where p is the vector of output probabilities, with each element corresponding to an output class. Suppose the set of all classes is represented by a vector C,
(2)$$\begin{array}{c} C={\left[C_{1},\;C_{2},\;\cdots ,\;C_{N}\right]}^{T}\in R^{N} \end{array}$$
where *N* is the total number of output classes and $C_{1},\;C_{2},\;\cdots ,\;C_{N}$ are the classes labels e.g., car, aeroplane, cat, dog, etc.

Since CNN outputs a probability for each class, the class with the highest probability is called the prediction of the neural network,
(3)$$\begin{array}{cc} i^{*} & =\underset{i}{\mathrm{argmax}}\;\;p\left[i\right],\\ C^{*} & =C[i^{*}], \end{array}$$
where ${\mathrm{argmax}}_{i}$ returns the index of the element of p with maximum value, [.] represent indexing operator into a vector and $C^{*}$ is the label of the predicted class.

Based on (Equation 1) and (Equation 3), it can be infered that the predicted class $C^{*}$ is actually a function of input image $X_{img}$, therefore we can write
(4)$$\begin{array}{c} C^{*}=F\left(X_{img}\right). \end{array}$$
where F(.) represent the combined effect of (Equation 1) and (Equation 3). Figure 3 shows the predictions of simple CNN architecture introduced in Figure 2 on real-world images.

For simplification of mathematical notation in later sections, let us define a new function P as follow
(5)$$\begin{array}{c} P\left(p,C_{i}\right)=p\left[i\right], \end{array}$$
which takes probability vector p and class name $C_{i}$ as input and outputs the probability of the corresponding class. Here [.] is the index operator and gives the *i*th element of vector p.

### 2.2. Image Perturbation

Our CNN fooling attack works by perturbing a finite number of pixels of the input image to change the output probabilities of the neural networks. In this paper, we used a *k*-pixel attack strategy in which only *k* pixels of the input image is allowed to be changed. Higher the value of *k*, easier it is to fool the CNN since we can just modify a large part of the image. The real challenge lies in successfully fooling the CNN using fewer pixels e.g., single-pixel (k=1) attack, where the attacking algorithm is only allowed to modify a single pixel.

To mathematically define the image perturbation function P, consider the input image $X_{img}$ has m×n rgb-pixels. The pixel intensity varies from 0 to 1 i.e., rgb∈[0,1]. The perturbation function takes three metrices with *k* numbers of rows as input; rows $r\in {\left[1,2,\cdots ,m\right]}^{k\times 1}$, columns $c\in {\left[1,2,\cdots ,n\right]}^{k\times 1}$ and rgb values $rgb\in {\left[0,1\right]}^{k\times 3}$. The function P returns a perturbed image $X_{per}$ such that the rows and columns provided in metrices *r* and *c* and changed with corresponding values in rgb matrix
(6)$$\begin{array}{c} X_{per}=P\left(X_{img},\;r,\;c,\;rgb\right). \end{array}$$

The working of perturbation function P is defined as follow:(7)$$\begin{array}{c} P:X_{img}\left[r\left[i\right],r\left[j\right]\right]:=rgb\left[i\right],\;\mathrm{where}\;i\in \left\{1,2,\cdots ,k\right\} \end{array}$$
where := is the assignment operator. Figure 4 shows an illustration of *k*-pixel attack.

The objective of a CNN fooling attack is to find a perturbed image $X_{per}$ such that the CNN outputs a wrong predicted class $C^{*}\neq C_{real}$. Two different types of attacks have been proposed; untargeted and targeted attacks. Now we will model each type of these attacks as an optimization problem.

### 2.3. Untargeted Attack

In an untargeted attack, the objective is as follows; to change the input image $X_{img}$ to a perturbed image $X_{per}$ such that the predicted class $C^{*}$ is not equal to the real class $C_{real}$. In an untargeted attack, we are not concerned about the value of the new predicted class. In other words, we just want to minimize the confidence of the CNN in the real class $C_{real}$. By using the definition from (Equation 5) and (Equation 1) we can write the following minimization problem
$$\begin{array}{c} X_{per}^{*}=\underset{X_{per}}{\mathrm{argmin}}\;\;P\left(f_{cnn}\left(X_{per}\right),C_{real}\right), \end{array}$$
where $X^{*}$ is the modified image which minimize the confidence of the CNN in the real class $C_{real}$. For k-pixel attack, (Equation 6) can be used to rewrite the above objective function as
(8)$$\begin{array}{cc} & r^{*},c^{*},rgb^{*}=\underset{r,c,rgb}{\mathrm{argmin}}\;\;P\left(f_{cnn}\left(P\left(X_{img},r,c,rgb\right)\right),\;C_{real}\right)\\ & \mathrm{Subject}\;\mathrm{to}:\\ & \;\;0\leq r[i,1]\leq m,\;\;\mathrm{where}\;\;i\in \{1,2,\cdots ,k\}\\ & \;\;0\leq c[i,1]\leq n,\;\;\mathrm{where}\;\;i\in \{1,2,\cdots ,k\}\\ & \;\;0\leq rgb[i,j]\leq n,\;\;\mathrm{where}\;\;i\in \{1,2,\cdots ,k\},\;j\in \{1,2,3\} \end{array}$$
where notation [i,j] represents indexing into a matrix and returns element at *i*th row and *k*th column. $r^{*}$, $c^{*}$ and $rgb^{*}$ are the required rows, columns and rgb values.

### 2.4. Targeted Attack

In a targeted attack, the objective of the attack is not just to change the output of the network from real class $C_{real}$, but also fool the network to output a target class $C_{target}\neq C_{real}$. The targeted attack adds a secondary requirement on the success criteria. To model it as an optimization problem, we need to use the fact that our objective is to increase the confidence of the CNN in the target class $C_{target}$. We can model it as following maximization problem.
$$\begin{array}{c} X_{per}^{*}=\underset{X_{per}}{\mathrm{argmax}}\;\;P\left(f_{cnn}\left(X_{per}\right),C_{target}\right), \end{array}$$
which is equivalent to the following problem in term of *k*-pixel attack,
(9)$$\begin{array}{cc} & r^{*},c^{*},rgb^{*}=\underset{r,c,rgb}{\mathrm{argmax}}\;\;P\left(f_{cnn}\left(P\left(X_{img},r,c,rgb\right)\right),\;C_{target}\right)\\ & \mathrm{Subject}\;\mathrm{to}:\\ & \;\;0\leq r[i,1]\leq m,\;\;\mathrm{where}\;\;i\in \{1,2,\cdots ,k\}\\ & \;\;0\leq c[i,1]\leq n,\;\;\mathrm{where}\;\;i\in \{1,2,\cdots ,k\}\\ & \;\;0\leq rgb[i,j]\leq 1,\;\;\mathrm{where}\;\;i\in \{1,2,\cdots ,k\},\;j\in \{1,2,3\} \end{array}$$

To use the same optimization framework for targeted and untargeted attacks, we can convert the above maximization problem to the following equivalent optimization problem,
(10)$$\begin{array}{cc} & r^{*},c^{*},rgb^{*}=\underset{r,c,rgb}{\mathrm{argmin}}\;\;1-P\left(f_{cnn}\left(P\left(X_{img},r,c,rgb\right)\right),\;C_{target}\right), \end{array}$$
where $r^{*}$, $c^{*}$ and $rgb^{*}$ are the rows, columns and rgb to change the output of the CNN to the targeted class.

## 3. Algorithm

In this section, we will present the algorithm to efficiently solve the optimization problem formulated in Section 2. First, we characterize the food foraging behavior of beetles and then mathematically formulate the algorithm.

### 3.1. Mathematically Modelling Behavior of Beetle

To formulate a computationally efficient optimization algorithm, requiring just a single search particle, we propose a nature-inspired optimization algorithm based on the food foraging behavior of beetles. It is in contrast to the traditional CNN fooling algorithms that require several search particles to fool the CNN. The Beetles have antennae-like structure attached to their heads. The antennae help them probe for the smell of food in the environment. This ability of using antennae to search for food is particularly interesting because it can efficiently explore an unknown environment and find the goal, i.e., food source, without any advanced sensory capabilities.

The problem mentioned above of searching for an food source is essentially an optimization problem. The distance from the food source is equivalent to the objective function. The goal of a beetle is to minimize the distance between itself and the food source. A beetle starts from a random location in the environment. At each step during the search, it uses antennae to probe its surrounding locations, i.e., calculate the value of the objective function at the location of antennae. The beetle estimate this value using the intensity of smell. Based on the probed values, it determines an incrementally favorable search direction toward the food source. It keeps on taking steps using the same strategy until finally reaching the optimal point of the objective function, i.e., food source. Figure 5 illustrates the concept.

### 3.2. Optimization Algorithm

To mathematically model the behavior of a beetle, let us consider the searching for the food source (i.e., place with maximum intensity of food smell) as an optimization problem. The map of smell intensity in the environment corresponds to the value of objective functions. The goal for beetle is to find the maxima of the smell intensity i.e., the food source. Let g(x) is the function representing the smell intensity at point x. The searching for maximum smell intensity is equivalent to the solution of following optimization problem with linear inequality constraints
(11)$$\begin{array}{cc} & \max_{x}\;g\left(x\right)\\ & \mathrm{Subject}\;\mathrm{to}:\;\;x_{\min}\leq x\leq x_{\max}. \end{array}$$

Suppose at time instant *t*, the beetle finds itself at position $x_{t}$. The intensity of smell is given by $g(x_{t})$, at the current location. In order to take the next step, the beetle measure intensity of smell in each direction using each of its antennae. Suppose the antennare is located in the radially opposite direction, and randomly generated $\vec{b}$ represents the direction vector of left antennae relative to the current position of the beetle $x_{t}$. The following equations give the position of antennae endpoints
(12)$$\begin{array}{cc} x_{l} & =x_{t}+\lambda \vec{b},\\ x_{r} & =x_{t}-\lambda \vec{b}, \end{array}$$
where λ is the length of an antennae, $x^{l}$, and $x^{r}$ are the position vectors of left and right antennae respectively. However, these vectors may violate the constraint of (Equation 11) because of randomly generated vector $\vec{b}$. Therefore, we define a constrained set Ψ as follow
$$\begin{array}{c} \Psi =\{x|x_{\min}<x<x_{\max}\}. \end{array}$$
and project the vectors $x_{l}$ and $x_{r}$ on the constrained set Ψ
(13)$$\begin{array}{c} {}^{\Psi }x_{l}=P_{\Psi }\left(x_{l}\right),{\;}^{\Psi }x_{r}=P_{\Psi }\left(x_{r}\right), \end{array}$$

Ψ is written in superscript to denote that the vectors are projected on set Ψ. The function $P_{\Psi }\left(.\right)$ is called a projection function. We defined it as follow,
(14)$$\begin{array}{c} P_{\Psi }\left(x\right)=\max\left\{x_{\min},\min\left\{x,x_{\max}\right\}\right\}. \end{array}$$

Such a definition of projection function is simple and and computational efficient.

The smell intensities at projected antennae location is given by $g{(}^{\Psi }x_{l})$ and $g{(}^{\Psi }x_{r})$. By comparing these values, the beetle take next step according to the following rule,
(15)$$\begin{array}{c} x_{new}=x_{t}+\delta \;\mathrm{sign}\left(g\left(x_{l}\right)-g\left(x_{r}\right)\right)\vec{b}, \end{array}$$
where the signum function sign(.) ensures that the next step is taken toward the direction of higher smell intensity. δ is the actual step-size taken by the beetle and proportional to Euclidean distance between $x_{new}$ and $x_{t}$. After reaching the new location the beetle will re-measure the intensity of smell; if there is an improvement it will remain at the new location; otherwise, it will return to the previous location, i.e.,
(16)$$\begin{array}{c} x_{t+1}=\left\{\begin{array}{cc} x_{new},\;\; & \mathrm{if},\;\;g\left(x_{new}\right)\geq g\left(x_{t}\right)\\ x_{k},\;\; & \mathrm{if},\;\;g\left(x_{new}\right)<g\left(x_{t}\right) \end{array}\right.. \end{array}$$

After reaching $x_{t+1}$, we again generate a random direction vector $\vec{b}$ and repeat the same process until reaching the goal. Although the above algorithm is formulated for the maximization problem (Equation 11), it can be converted used for the minimization problem by modifying the update rule in (Equation 15) as
(17)$$\begin{array}{c} x_{new}=x_{t}-\delta \;\mathrm{sign}\left(g\left(x_{l}\right)-g\left(x_{r}\right)\right)\vec{b}. \end{array}$$

The algorithm can be summarised as following:Start from random location $x_{0}$.Generate a random direction vector $\vec{b}$ for left antennae relative to current position $x_{0}$ of the beetle.Calculate the position of left and right antennae ($x^{l}$ and $x^{r}$) using (Equation 12).Calculate new position $x_{t+1}$ using (Equation 15) and (Equation 16).If reached goal position $x_{G}$, stop. Otherwise, return to step 2.

To use this algorithm for solving the optimization problem of untargeted and targeted attack defined in (Equation 8) and (Equation 10) respectively, we define a matrix $\bar{X}$
(18)$$\begin{array}{c} \bar{X}=\left[\begin{array}{ccc} r & c & rgb \end{array}\right]. \end{array}$$

By considering the notation defined in Section 2.2, the dimension of $\bar{X}$ becomes k×5. This definition of a new matrix $\bar{X}$ allow us to define the objective function using a single variable. For an untargeted attack, the objective function becomes,
(19)$$\begin{array}{c} g\left(\bar{X}\right)=P\left(f_{cnn}\left(P\left(X_{img},\bar{X}\left[:,1\right],\bar{X}\left[:,2\right],\bar{X}\left[:,\left\{3,4,5\right\}\right]\right)\right),\;C_{real}\right), \end{array}$$
where the semicolon symbol (:) in the matrix indexing is used to denote the entire column of the matrix. Similarly, for the targeted attack, the objective function becomes,
(20)$$\begin{array}{cc} & g\left(\bar{X}\right)=1-P\left(f_{cnn}\left(P\left(X_{img},\bar{X}\left[:,1\right],\bar{X}\left[:,2\right],\bar{X}\left[:,\left\{3,4,5\right\}\right]\right)\right),\;C_{target}\right). \end{array}$$

Since *r* and *c* in (Equation 18) can only take integer values, we use the round function to convert floating-point values to integers. Based on these mathematical relations for the objective function of untargeted and targeted attacks, we formally present the steps of optimization algorithm in Algorithm 1.

### 3.3. Illustration of Attacking Algorithm

Figure 6 illustrates the proposed attacking strategy. The CNN model takes a 32×32×3 RGB image as input and outputs the class of objects present in the image, i.e., horse, dog, car, etc. For the demonstration of the fooling algorithm, we used bettle with just two antennae. The original input image contains a horse. Figure 6a shows the CNN correctly predicts the class to be a horse for the original with the confidence of 95.7%. Then we start the iterations of our fooling algorithm by initializing the algorithm at a random pixel location. Figure 6b shows the modified image after the first iteration; the modified pixel is highlighted in red. At this iteration, CNN still predicts the correct class, but the confidence of reduced to 93.67%. Figure 6c shows further decrease in confidence after a mere 32 iterations. Finally, the beetle is able to find a pixel to fool the CNN after 200 iterations, as shown in Figure 6d. The network is fooled into predicting that the input image contains a cat, whereas, in reality, only one pixel of the input image is modified.

### 3.4. Computational Complexity

The computational complexity of the proposed algorithm can be computed by analyzing the steps listed inside while loop of Algorithm 1. The first step of the algorithm requires the generation of (3k+k+k=)5k uniformly distributed random variables, where *k* denotes the number of attacked pixels. On modern processors with the native ability to generate random numbers, it will require 5k operations in the worst case (support for vectorized operation will considerably reduce this number). The second step, i.e., calculation of antennae’s end-point locations, requires (2×5k=)10k multiplication and a similar number of additions, making up a total of 20k floating-point operations in this step. The next step requires the evaluation of the objective function twice. A careful analysis of the objective function tells that it basically consists of two primary steps: (1) perturbing the input image and (2) forward passing the perturbed image through the CNN. Since the neural networks are very computationally expensive systems and even a single forward pass can require a large number of floating-point operations. Although the exact numbers of floating-point operations depend on the design of CNN architecture, even small networks, such as LeNet, can require millions of computations. Let’s suppose the number of floating-point operations required by the CNN model are *N*. Since we need to forward pass through the CNN model twice, which brings the number of floating-point operations 2N. Adding all computations for all these steps, we reach a final value of 5k+20k+2N≈2N. The last approximation is based on the fact that N>>k, i.e., the number of computations required by CNN, is much higher than the number of attacked pixels. From this analysis, it can be seen that the amount of computation per iteration is primarily dominated by the CNN model and the computation requirement of BAS are negligible in comparison.

**Algorithm 1:** Attacking Algorithm.

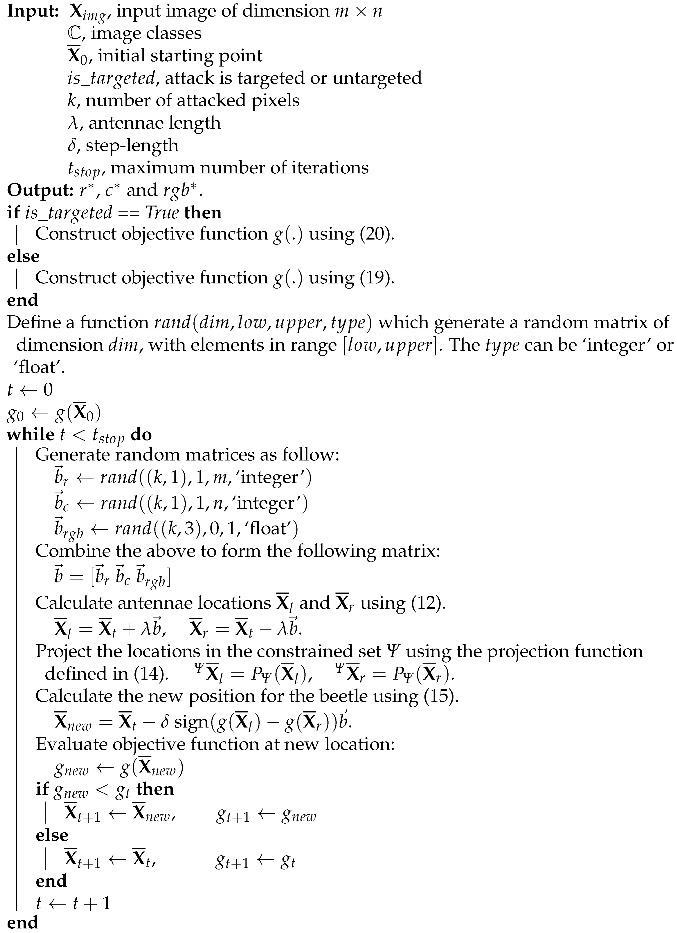



## 4. Experiment Methodology and Results

In this section, we will present the experimental methodology to evaluate the success rate of the proposed attacking algorithm. Then we will discuss the results.

### 4.1. Evaluation Methodology

To statistically evaluate the effectiveness of the proposed algorithm we choose two CNN architectures; LeNet-5 [63] and ResNet [64], trained on CIFAR-10 dataset [65].

#### 4.1.1. Image Dataset

The CIFAR-10 dataset consists of a total of 60,000 RGB images. Each image have a dimension of 32×32×3. The images belong to 10 classes, namely airplane, automobile, bird, cat, deer, dog, frog, horse, ship, truck. The dataset is evenly distributed among ten classes i.e., 6000 images belong to each class. The dataset is divided into two portions; the first portion is a training dataset, contains 50,000 images, while the second portion contains 10,000 images reserved for testing. The dataset is widely used in the training and testing of computer vision and machine learning models. The dataset is split into 50,000 training and 10,000 test images. Figure 7 shows one sample image from each of the ten classes of the dataset.

#### 4.1.2. LeNet-5 Architecture

LeNet is a five-layer CNN architecture. It consists of two 2D-convolutional with downsampling layers, followed by two fully connected layers and the softmax output layer. The architecture of LeNet is shown in Figure 8a. The input to the LeNet is a 32×32×3 RGB image from CIFAR-10 dataset and output is a 10×1 probability vector. The hidden layers of LeNet-5 are in the following order.
The first hidden layer of LeNet is a 2D convolutional layer with six kernels, each of dimension 5×five×3. Each kernel uses a rectified linear unit (ReLU) as an activation function. The total tunable parameter in this layer, including the bias parameters, are 5×5×3×6+6=456. A max-pooling layer follows the convolutional layer with a stride of (2,2).The second hidden layer is similar 2D convolutional layer with 16 kernels, each of dimension 5×5×6. Total number of tunable parameters in this layer are 5×5×6×16+16=2416.The output of the second convolutional layer is flattened from a 5×5×16 to a 400×1 vector. The flattened layer is connected to a fully connected later with 120 neurons with ReLU activation. The total trainable parameters in this layer are 400×120+120= 48,120.The fourth layer is also a fully connected layer with a total of 84 neurons with ReLU activation. Connection with layer three makes the total trainable parameters in this layer to be 120×84+84= 10,164.The last layer is a fully-connected layer with ten neurons using softmax activation. The connection with fourth layer makes a total of 84×10+10=850 trainable parameters. The output is 10×1 vector.

The above mentioned 5 layers make the total trainable parameter count for LeNet-5 architecture to be 62,006.

#### 4.1.3. ResNet Architecture

Residual Network (ResNet) architecture is another popular CNN architecture that employs the concept of residual learning to train the neural networks efficiently. The used ResNet architecture is shown in Figure 8b. It can be seen that the architecture contains a special type of block called residual blocks. Each residual block contains two 2D-convolutional layers. Apart from the usual forward path through convolutional layers, the residual blocks also have an alternate shortcut forward path. This shortcut path allows the convolutional layers only to learn residual mapping instead of actual mapping. There are a total of 15 residual blocks and 15×2+1=31 convolutional layers. Similar to LeNet-5 architecture, the ResNet takes a 32×32×3 RGB image as input and output a probability vector. Some features of ResNet architecture are briefly given below.
The first hidden layer is a 2D convolutional layer with 16 kernels, each of dimension 3×3×3. This convolutional layer uses zero paddings; therefore, the output of this layer has a dimension of 32×32×16.The first convolutional layer is followed by a set of 5 similar residual blocks. Each residual block contains two convolutional layers. Each of the convolutional layers contains 16 kernels of dimension 3×3 and uses zero paddings to maintain the dimension between its input and output. Each convolutional layers outputs a 3D-matrix of dimension 32×32×16, except the output of fifth residual block which apply max polling with stride (2,2), making output dimension 16×16×16.After the fifth residual block, we have another set of 5 residual blocks. For this set of the block, each convolutional layer has a total of 32 kernels of dimension (3×3). Each convolutional layer outputs a 3D-matrix of dimension 16×16×32, except the output of the tenth block, which employs a max-pooling layer and outputs a 3D-matrix of dimension 8×8×32.The residual blocks from eleventh to fifteenth are also similar to each other. The convolutional layers in these blocks have a total of 64 kernels of dimension (3,3). The output of each convolutional layers is a 3D-matrix of dimension 8×8×64.The output of the fifteenth residual block is passed through a global average pooling layer and outputs a 3D-matrix of dimension 8×8×1.The 8×8×1 tensor is flattened into 64×1 vector and connected with a fully connected layer of 10 neurons with softmax activation.

Apart from the output layer, all the remaining layers use ReLU activation. The total number of trainable parameters in this ResNet architecture are 467,946. This number is almost 7.5 times larger then LeNet-5. Due to the higher number of parameters and more complexity, the ResNet architecture shows much better performance as compared to LeNet-5.

#### 4.1.4. Training of CNNs

We implemented both CNN architectures in TensorFlow [66] and trained them using the 50,000 training images from the CIFAR-10 dataset using categorical cross-entropy as the loss function. Figure 8c shows the prediction of the CNN for some sample images along with the corresponding confidence values for the trained LeNet-5 and ResNet models. It can be seen that the confidence value of the ResNet model is comparatively higher than the LeNet-5 model for most of the image. It can be attributed to the larger size and complex structure of ResNet. The complex structure allows the ResNet to learn more image features as compared to LeNet-5. The prediction accuracy of the LeNet-5 and ResNet model on the training and testing models are summarized in Table 2.

### 4.2. Results and Discussion

To conduct a fair evaluation of our proposed algorithm, we only attacked images from the test dataset, which are correctly classified by the corresponding trained models. According to Table 2, only 7488 out of 10,000 test images were used in fooling attack on LeNet-5. Similarly, for ResNet, 9231 out of 10,000 test images were considered. Now we will present the results for untargeted and targeted attacks on both of these networks separately.

Figure 9 shows some of the sample images for untargeted fooling attacks. Figure 9a corresponds to the images for the LeNet-5 and Figure 9b present the images for ResNet. Each image also mentions the pre-attack prediction of the corresponding CNN along with the confidence. The post-attack prediction of the wrong class, along with the new confidence value, is also shown. The modified pixel is shown as red. The number of iterations required to find the pixel to fool the CNN is also shown. An interesting observation is to note that the CNN can be fooled even when the perturbed pixel does not directly lie on the actual object.

Similarly, Figure 10 shows the sample images for targeted fooling attacks. Figure 10a corresponds to the images for the LeNet-5 and Figure 10b present the images for ResNet. In addition to the information described above for Figure 9, Figure 10 also shows the target class. The target class for each image is chosen randomly, however it was made sure that the it is never same as the real class of the image.

To compile the statistical results, we randomly selected a total of 300 images. We then used a 5-fold attacking method, i.e., each image was attacked five times. We also evaluated the performance of the proposed algorithm for the case of 3-pixel and 5-pixel attacks. The distribution of success rate is shown as a histogram and pie chart in Figure 11. Figure 11a shows the success rate distribution for the attacks on LeNet-5 architecture. The left-most end of the histogram represents the images on which all attempts failed. Whereas, the right-most end contains the images with success on all five trials. Although most samples are contained at the edges of the histogram, still there are some samples in the middle. It indicates that the algorithm shows probabilistic behavior, i.e., succeed and failed on the same input image. Additionally, it can be seen that for higher number of pixels, the success rate also increases. It is intuitive since the more the number of perturbed pixels, the greater is the difference between the original image and the new image. Similarly, it can be seen that the success rate for the untargeted attack is higher as compared to the targeted attack.

We repeated the same set of experiments for the ResNet. As expected, the attack on ResNet proved to be difficult as compared to LeNet and achieved a lower success rate. Figure 11c summarize the success rate for targeted and untargeted attack on ResNet for the case of 1, 3 and 5-pixel attacks on ResNet. As an overall summary, Figure 11b shows the distribution of the number of success in all the 5-fold attacks. It shows that the proposed algorithm is able to fool the CNNs for a large proportion of images successfully.

## 5. Conclusions

Experimental results presented in this paper clearly show that, from a computational point of view, the behavior of an rudimentary-intelligent insect, i.e., beetle, is fully capable of fooling artificially intelligent computation models. These results can help provide insight into the nature of artificial intelligent learning systems and their comparison with natural intelligence. Additionally, this paper emphasizes the need for robustness in the designing of artificial learning systems. We demonstrate the efficacy of the proposed CNN attacking algorithm by considering two CNN architectures; LeNet-5 and ResNet, trained on the CIFAR-10 dataset. The statistical results show that the algorithm was able to successfully fool CNN for a large proportion of the input images.

## Figures and Tables

**Figure 1 biomimetics-07-00084-f001:**
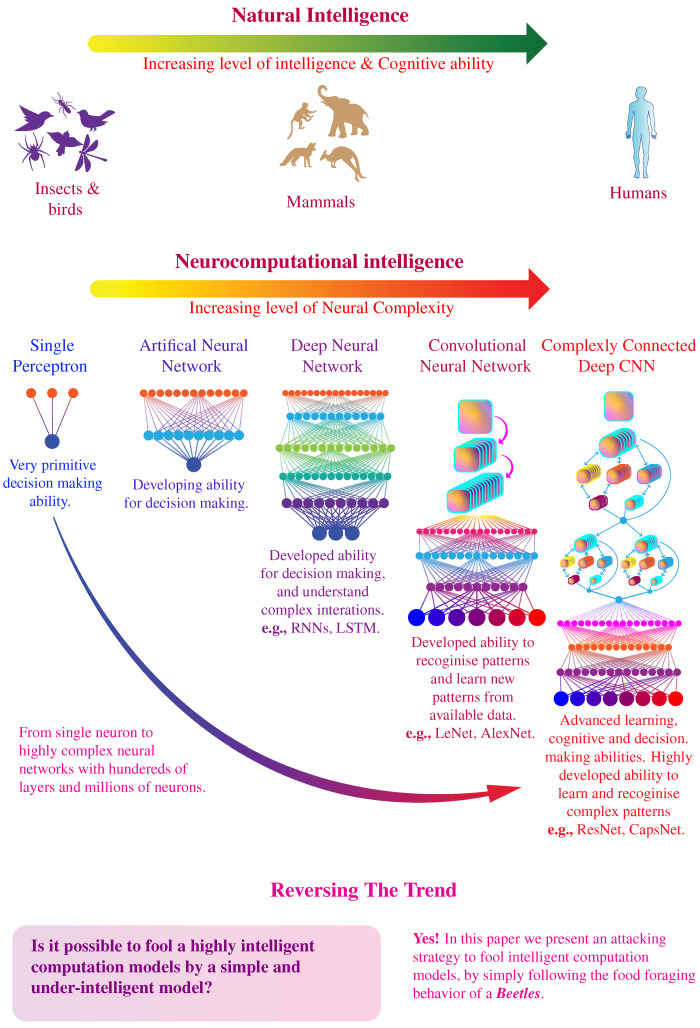
Evolution of artifical and neurocomputational intelligence. The spectrum of natural intelligence ranges from simple organisms to highly intelligent human brains. The spectrum of artificial neurocomputational intelligence also follows a similar trend in term of complexity and ranges from single perceptron to multi-layered neural networks comprising of millions of neurons.

**Figure 2 biomimetics-07-00084-f002:**
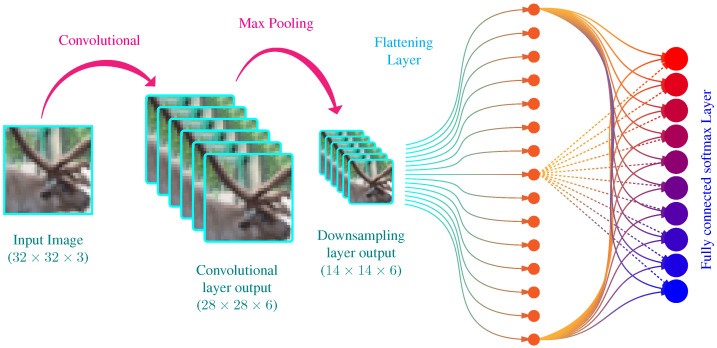
A simple structure of CNN for an image classification task. The network consists of one convolutional layer, one downsampling layer, and a fully connected layer followed by the output layer. The network output the probabilities that the image contains particular objects.

**Figure 3 biomimetics-07-00084-f003:**
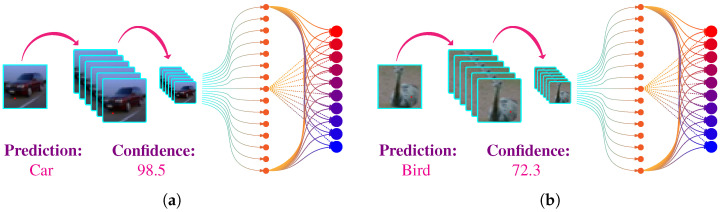
Prediction of a trained simple CNN network on real-world images. (**a**) The network predict that the input image is a car with a confidence of 98.5%. (**b**) The network predict that the input image is a car with a bird of 98.5%.

**Figure 4 biomimetics-07-00084-f004:**
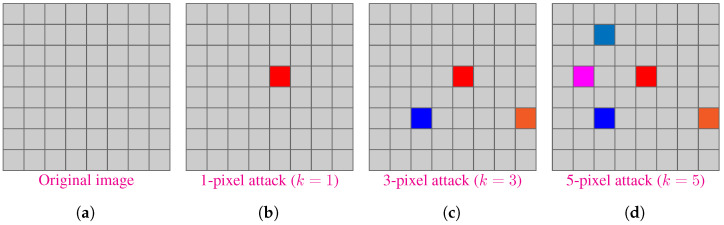
Illustration of *k*-pixel perturbation technique for attacking a CNN. It shows a image with a grid of 8×8 pixels. (**a**) Original Image, (**b**) k=1, (**c**) k=3, and (**d**) k=5. A pixel is assigned a random RGB value for attack.

**Figure 5 biomimetics-07-00084-f005:**
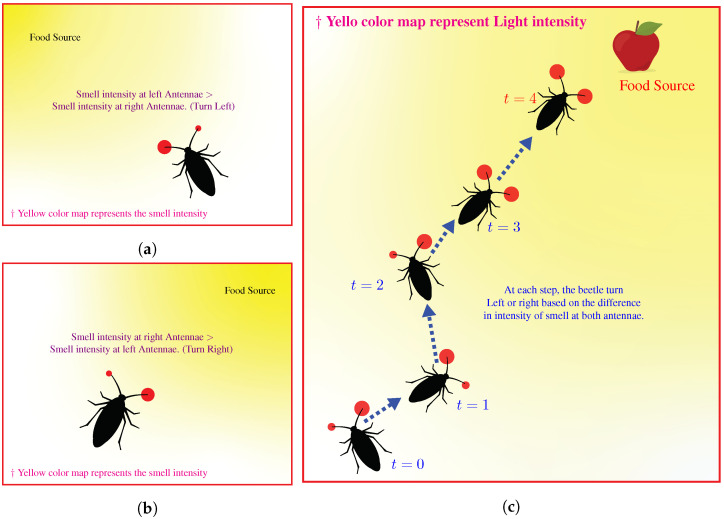
Illustration of Beetle’s strategy during searching for food source, i.e., maximum intensity of food smell. (**a**,**b**) shows different configuration of food source and the beetle. (**c**) Illustration of the locomotion of the microorganism.

**Figure 6 biomimetics-07-00084-f006:**
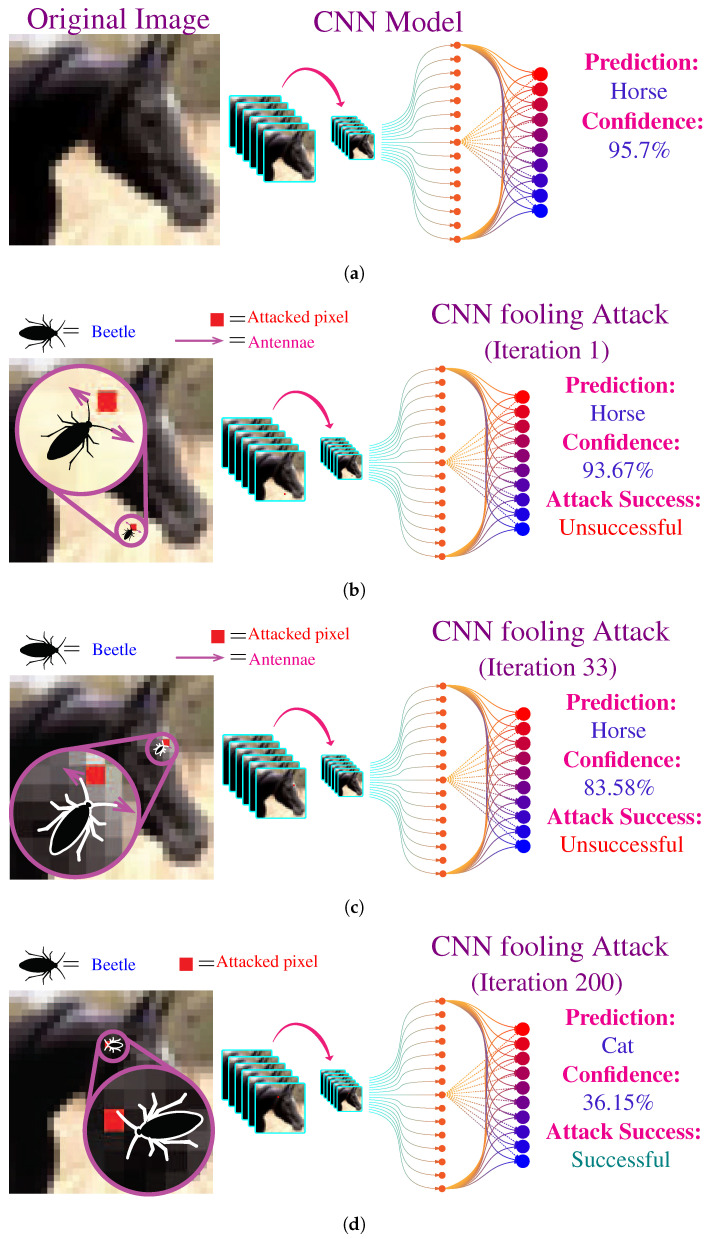
Illustration of the CNN attacking algorithm. (**a**), The original input image to the CNN model: the image contains a horse, and the CNN’s prediction is also a horse with the confidence of 95.7%. (**b**), First iteration of the fooling attack: the modified image reduced the confidence to 93.67% with just modification of single pixel. (**c**), 33rd iteration of the fooling attack: after searching for a while, the algorithm found a pixel which reduced confidence to 83.58%. (**d**), 200th iteration of the fooling attack: the algorithm finally found a pixel which fools the CNN to classify the image as a cat.

**Figure 7 biomimetics-07-00084-f007:**
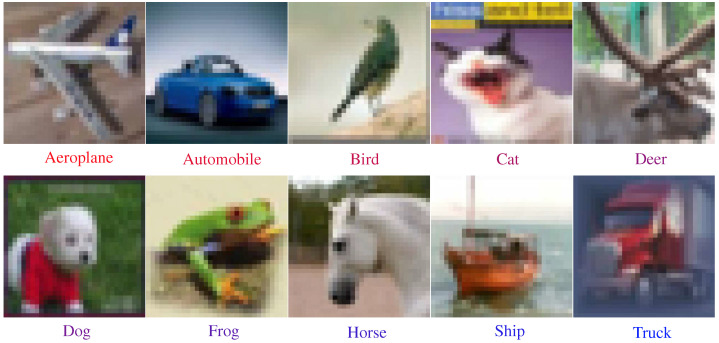
Sample images from the CIFAR-10 dataset. The dataset contains a total of 10 classes, including animals and objects. One image from each class is shown here.

**Figure 8 biomimetics-07-00084-f008:**
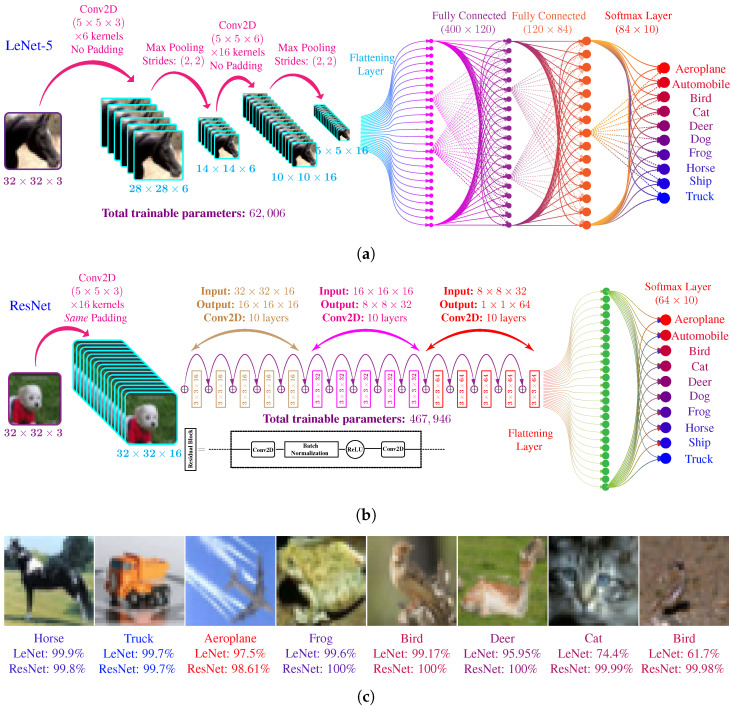
CNN architecture used for experiments in this paper. (**a**) Topology of LeNet-5, it contains a total of 5 trainable layers. (**b**) Topology of ResNet, it contains a total of 32 trainable layers. (**c**) Prediction of trained LeNet-5 and ResNet models for sample images of CIFAR-10 dataset. ResNet usually show high confidence as compared to LeNet-5 because of higher number of layers.

**Figure 9 biomimetics-07-00084-f009:**
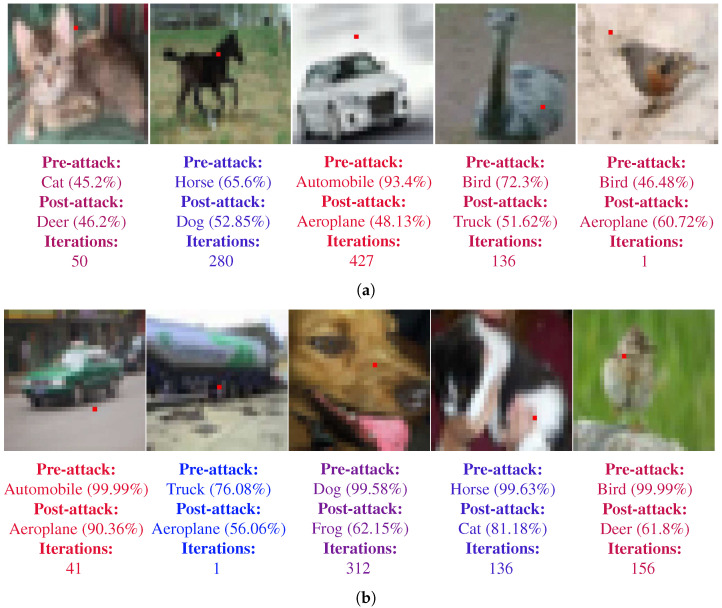
Samples of single pixel successful untargeted attacks on LeNet-5 and ResNet architectures. Perturbed pixel is marked as red. The pre-attack, post-attack predictions and confidences along with the number of iteration required to fool the CNN are also shown below the images. (**a**) Samples for LeNet-5. (**b**) Samples for ResNet.

**Figure 10 biomimetics-07-00084-f010:**
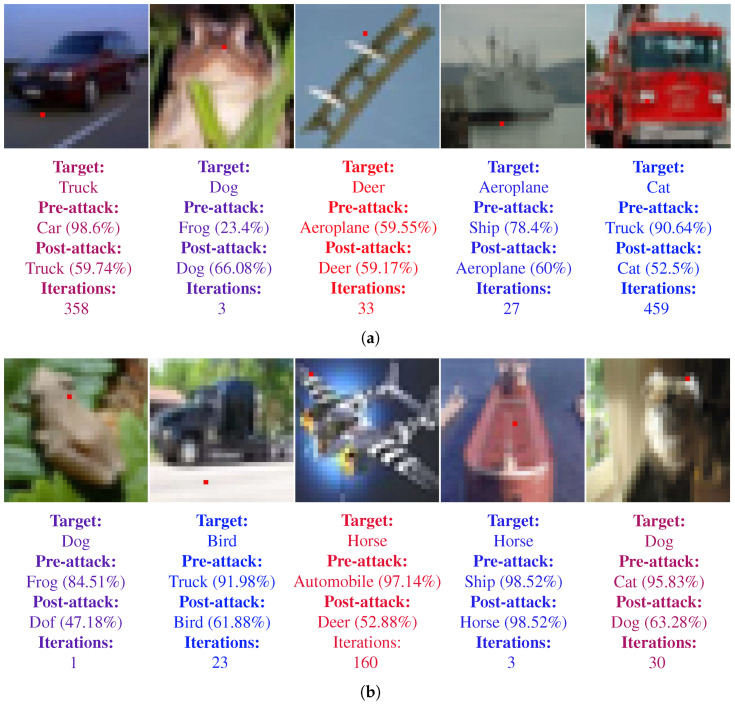
Samples of single pixel successful targeted attacks on both LeNet-5 and ResNet architectures. Perturbed pixel is marked as red. The pre-attack, post-attack predictions and confidences along with the number of iteration required to fool the CNN are also shown below the images. (**a**) Samples for LeNet-5. (**b**) Samples for ResNet.

**Figure 11 biomimetics-07-00084-f011:**
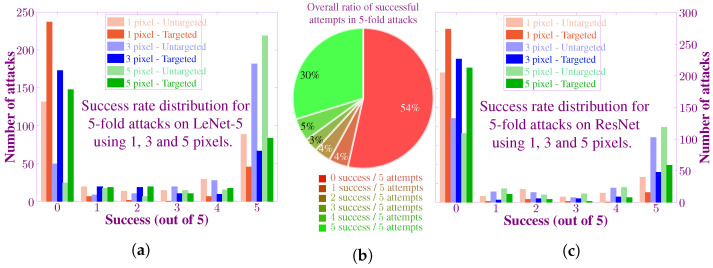
Statistics of the attack success rate for all experimental scenarios. (**a**) The histogram shows the success rate distribution in 5-fold attacking experiments for LeNet-5. Random images from the CIFAR-10 test dataset are attacked five times each using 1,3 and 5-pixels, and the number of successful attempts is recorded as shown in the histogram. The targeted attacks have a lower success rate as compared to untargeted attacks. Additionally, as the number of attacked pixels increases, the success rate also improves. (**b**) The proportion of successful and unsuccessful attacks for all of the experiments. (**c**) Success rate distribution for ResNet in both targeted and untargeted attacks.

**Table 1 biomimetics-07-00084-t001:** Comparison between metaheuristic algorithms proposed in the literature for fooling Neural Network Models.

Algorithm	Nature-Inspired	Attacked Model	Dataset Type	Number Search Particles
Grey wolf optimization [38]	Yes	AlexNet	Image sequences	Several
SIGMA [39]	No	Neural Networks	Network Intrusion detection dataset	30
PSO [40]	Yes	BiLSTM and BERT	Text (Natural Language)	8
Differential Evolution [41]	No	VGG16 and AlexNet	Images (CIFAR-10)	400
BAS (proposed)	Yes	LeNet and ResNet	Images (CIFAR-10)	2

**Table 2 biomimetics-07-00084-t002:** Summary of trained CNN architectures.

×	No. of Parameters	Training Samples	Training Accuracy	Testing Samples	Testing Accuracy
LeNet-5	50,000	50,000	78.47%	10,000	74.88%
ResNet	50,000	50,000	99.83%	10,000	92.31%

## Data Availability

Not applicable.

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
