# Peer review of "Beetle Antennae Search: Using Biomimetic Foraging Behaviour of Beetles to Fool a Well-Trained Neuro-Intelligent System"

_biomimetics, 2022, doi:10.3390/biomimetics7030084_

Round 1
Reviewer 1 Report
In the reviewed paper, the authors focused their research on analyzing attacks on neural network models. Especially on CNN-based architectures. It is an important topic-neutral network due to the expansion of federated learning ideas. Poisoning and other different attacks can change the whole training model. Therefore, this research is important and very interesting. I have only a few comments:
1. Explain in more detail why did you use such a heuristic algorithm when there are so many other ones (red fox, black widow, etc). Some discussion and comparison are needed.
2. Explain your solution of using a heuristic to other known, similar ideas poisoning dataset attack providers in meta-heuristic as a manager in federated learning.
3. Time/computational complexities also should be discussed.
4. Evaluate the impact of training iteration number and population size on the fooling CNN model.
5. Some comparison with state-of-art should be added.
Author Response
Please refer to the "Responses to Comments of Reviewer #1" section in the attached file.

Reviewer 2 Report
In general paper is interesting, not complex but clear presentation of simple idea. Improve presentation of novelty in your approach and new features of this idea. Also add comparisons to other models so that we can better see your advances.
Author Response
Please refer to the "Responses to Comments of Reviewer #2" section in the attached file.

Round 2
Reviewer 1 Report
The paper was improved and can be accepted in the current version.